# Molecular architecture of fungal cell walls revealed by solid-state NMR

Xue Kang [1], Alex Kirui [1], Artur Muszyński [2], Malitha C. Dickwella Widanage [1], Adrian Chen[1], Parastoo Azadi[2], Ping Wang [3], Frederic Mentink-Vigier [4] & Tuo Wang [1]

The high mortality of invasive fungal infections, and the limited number and inefficacy of antifungals necessitate the development of new agents with novel mechanisms and targets. The fungal cell wall is a promising target as it contains polysaccharides absent in humans, however, its molecular structure remains elusive. Here we report the architecture of the cell walls in the pathogenic fungus *Aspergillus fumigatus*. Solid-state NMR spectroscopy, assisted by dynamic nuclear polarization and glycosyl linkage analysis, reveals that chitin and α-1,3-glucan build a hydrophobic scaffold that is surrounded by a hydrated matrix of diversely linked β-glucans and capped by a dynamic layer of glycoproteins and α-1,3-glucan. The two-domain distribution of α-1,3-glucans signifies the dual functions of this molecule: contributing to cell wall rigidity and fungal virulence. This study provides a high-resolution model of fungal cell walls and serves as the basis for assessing drug response to promote the development of wall-targeted antifungals.

---

[1] Department of Chemistry, Louisiana State University, Baton Rouge, LA 70803, USA. [2] Complex Carbohydrate Research Center, University of Georgia, Athens, GA 30602, USA. [3] Departments of Pediatrics, and Microbiology, Immunology and Parasitology, Louisiana State University Health Sciences Center, New Orleans, LA 70112, USA. [4] National High Magnetic Field Laboratory, Tallahassee, FL 32310, USA. Correspondence and requests for materials should be addressed to T.W. (email: tuowang@lsu.edu)

Fungi are a group of eukaryotic microorganisms, some of which are capable of causing superficial infections or serious systemic diseases in humans. Superficial infections affect nearly a quarter of humans, but more importantly, invasive infections caused by fungi such as the unicellular *Candida* species and filamentous *Aspergillus fumigatus* often result in fatality in individuals with immunodeficiency. To date, life-threatening fungal infections affect more than two million people worldwide, with an exceptionally high-mortality rate of 20–95%[1]. As one of the most prevalent airborne fungi, *A. fumigatus* causes fatal invasive aspergillosis in more than 200,000 patients annually, including a quarter of all leukemia patients, with an overall mortality rate of 50% for patients with treatment and nearly 100% for those left untreated[2–6]. The high-mortality rate is also coupled with a substantial rise in occurrence due to a fast-growing population with immunodeficiency and the wide application of immunosuppressive agents in medical treatments such as cancer therapy or organ transplantation.

Despite the above described medical significance, effective antifungal agents remain very limited. Most available antifungals target ergosterols in the cell membrane and therefore are toxic to humans[7,8]. In addition, these antifungal drugs have limited efficacy. For example, Amphotericin B fails to prevent death in more than half of the patients with invasive aspergillosis[9]. Moreover, a substantial increase in drug resistance has been observed during the last decades[6,8]. Recent efforts have been devoted to developing agents that bind to the fungal cell wall since its polysaccharides are absent in human cells[10,11]. Echinocandins are such new compounds that disrupt glucan synthesis and perturb cell wall integrity with reduced toxicity[12–14]. However, echinocandins are not broad-spectrum drugs and are very expensive. All this makes it imperative to develop new compounds with better functional mechanisms or different primary targets such as the polysaccharides in the cell walls. One of the major challenges is that the fungal cell wall structure is poorly understood, placing a barrier to the development of cell wall-targeted antifungal agents.

Fungal cell walls typically contain, by weight, 50–60% glucans, 20–30% glycoproteins, and a small portion of chitin, for example, 10–20% for *A. fumigatus*[10,15]. Fungal glucans contain the predominantly linear β-1,3-linkage and a small portion, typically 10% or less, of β-1,6- and β-1,4-linkages. The supramolecular assembly of these biomolecules remains vague due to the lack of a non-destructive and high-resolution technique for characterizing the insoluble, complex, and amorphous biomolecules within the intact cell wall[16]. To date, chitin microfibrils are thought to be deposited next to the plasma membrane following the biosynthesis of individual chains and the fibril formation process through hydrogen bonding. These microfibrils may be covalently linked to β-1,3-glucans[17] that extend through the cell wall and tether mannoproteins on the cell wall surface through branched networks with β-1,6-linked glucans[18]. The current understanding of the spatial packing has been shaped by the evidence from enzymatic digestion, fractional solubilization, and isolation of cell wall components followed by sugar analysis[18,19]. These chemical and enzymatic methods, however, are destructive and often fail to reveal the complicated polymer assembly generated by biosynthesis machinery.

Recently, magic-angle spinning (MAS) solid-state NMR (ssNMR) spectroscopy has been employed to elucidate the structure, spatial proximities, and ligand binding of native or genetically modified polysaccharides in intact tissues, including the bacterial biofilm, plant biomass, and fungal pigment assemblies[20–28]. Here, by integrating glucosyl linkage analysis, 2D ¹³C–¹³C/¹⁵N correlation ssNMR spectroscopy and the sensitivity-enhancing dynamic nuclear polarization (DNP)[29–37] technique, we have successfully revealed the structural frame of the cell wall

in uniformly ¹³C, ¹⁵N-labeled *A. fumigatus*. We found that chitin and α-1,3-glucans closely interact to form a rigid and hydrophobic scaffold that is surrounded by a soft and well-hydrated matrix of β-glucans. Glycoproteins and a minor fraction of α-1,3-glucans form a highly dynamic shell coating the cell wall surface. In addition, we have revealed that fungal cell wall molecules adopt polymorphic structures and heterogeneous dynamics in order to perform versatile functions. Our findings also shed light onto the machinery and mechanisms of cell wall component biosynthesis and their assembly. The methods presented in this study enable the investigation of complex carbohydrates in intact cells and will allow the direct detection of fungal responses to antifungal agents through in-situ assessment of cell wall structures.

## Results

**Chitin and glucans form the stiff fungal cell wall.** Uniformly ¹³C, ¹⁵N-labeled *A. fumigatus* samples were grown in a ¹³C/¹⁵N liquid medium for 14 days. For ssNMR experiments, the *A. fumigatus* samples are analyzed in the intact and native state with minimal perturbation. We rely on the adequate sensitivity provided by isotope labeling and the resolution from a series of two-dimensional (2D) ¹³C–¹³C and ¹³C–¹⁵N correlation spectra for assigning NMR resonances and analyzing the composition, mobility, intermolecular packing and site-specific water interactions of these complex carbohydrates in muro.

Glycosyl compositional analysis, assisted by ssNMR, demonstrated a major component of glucan (71%), chitin (9%), mannan (6%), and galactan (13%), as well as traces of chitosan and arabinan in *A. fumigatus* (Supplementary Table 1). A gas chromatography–mass spectrometry (GC-MS) glycosyl linkage analysis of partially methylated alditol acetates (PMAA) (Supplementary Fig. 1) further revealed the highly diverse linkage patterns of fungal glucans. The major form, 3-linked glucopyranosyl (3-Glc*p*), accounts for 73% of all neutral sugars and 86% of Glc*p* residues, indicating the dominance of 1,3-glucans (Table 1). Another five types of Glc*p* linkages are also identified, comprising 11% of all neutral sugars. Since glucans are better solubilized in the linkage analysis than in the classical alditol acetate method of the compositional analysis, minor discrepancies between these two methods are possible.

To assign the NMR resonances of cell wall polysaccharides, we measured 2D ¹³C–¹³C correlation spectra using 53-ms CORD mixing[38,39] for through-space correlations (Fig. 1a) and refocused ¹³C INADEQUATE pulse sequence[40,41] for through-bond correlations (Fig. 1b). These 2D ¹³C–¹³C correlation experiments preferentially detect the stiff cell wall due to the use of ¹H–¹³C cross polarization (CP). The *A. fumigatus* cell wall exhibits exceptionally high resolution and the typical ¹³C full-width at half-maximum (FWHM) linewidths range from 0.4 to 0.7 ppm. Major signals are from chitin, β-1,3-glucan, and α-1,3-glucan (Fig. 1a,b). This is consistent with the dominance of 3-Glc*p* in the linkage analysis. The unique downfield ¹³C chemical shift of 80–87 ppm at the linkage site of carbon 3 (C3) resolves the signals of 1,3-glucans, and the C1 chemical shift tells the two anomers apart: 99–101 ppm and 102–105 ppm for α- and β-1,3-glucans, respectively. Weaker signals from β-1,4- and β-1,6-glucans have also been identified, with downfield ¹³C chemical shifts of 85 and 67 ppm at the linkage sites of C4 and C6, respectively. The low intensity of β-1,4- and β-1,6-glucans is also in good agreement with the glycosyl linkage analysis: only 7% of neutral sugars are glucans with linkages at C4 or C6 (Table 1). The representative structures are shown in Fig. 1c and the ¹³C and ¹⁵N chemical shifts are documented in Supplementary Table 2. These sugar units may be covalently linked to form branched structures such as the β-1,3/-1,6-glucan.

**Table 1 ¹³C-glycosyl linkages of fungal neutral sugars**

| Glycosyl residue | A. fumigatus | A. niger |
|---|---|---|
| Terminally linked mannopyranosyl (t-Man*p*) | 1.4 | 0.1 |
| Terminally linked glucopyranosyl (t-Glc*p*) | 2.6 | 2.5 |
| Terminally linked galactofuranosyl (t-Gal*f*) | 1.8 | 0.4 |
| 3-linked glucopyranosyl (3-Glc*p*) | 72.8 | 81.1 |
| 4-linked glucopyranosyl (4-Glc*p*) | 3.9 | 4.3 |
| 2,3-linked glucopyranosyl (2,3-Glc*p*) | 2.3 | 4.0 |
| 2,6-linked glucopyranosyl (2,6-Glc*p*) | 0.5 | 0.3 |
| 3,6-linked glucopyranosyl (3,6-Glc*p*) | 2.1 | 2.8 |
| 2-linked mannopyranosyl (2-Man*p*) | 1.7 | 0.8 |
| 6-linked mannopyranosyl (6-Man*p*) | 0.8 | 0.5 |
| 4-linked galactopyranosyl (4-Gal*p*) | 10.0 | 3.4 |

The percentages reported are peak area from the relative EI detector response (%). The intact cell walls of *A. fumigatus* and the alkali-insoluble portion of *A. niger* cell walls are reported.

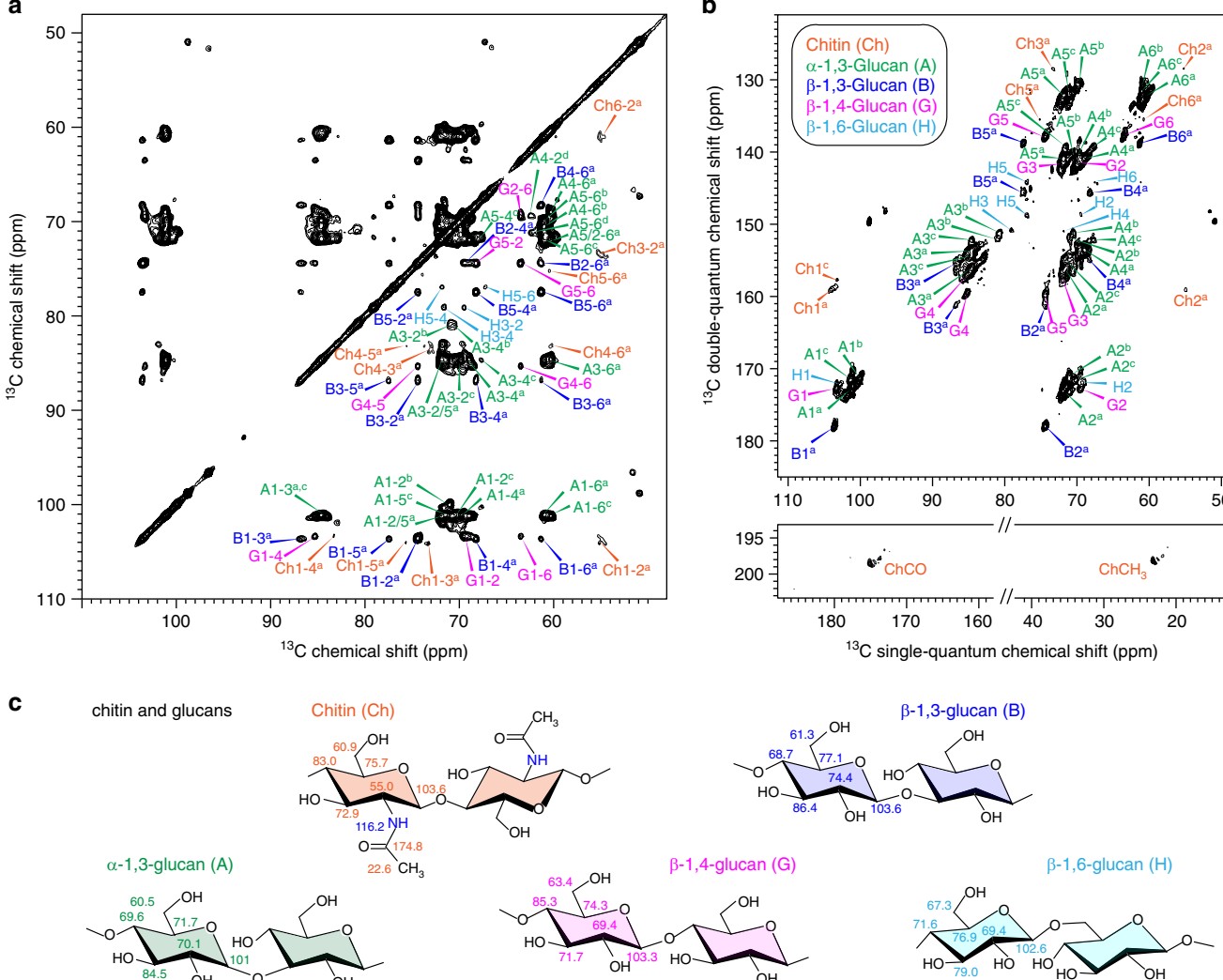

**Fig. 1** Chitin and glucans form the rigid domain of intact *A. fumigatus* cell walls. **a** 2D ¹³C–¹³C correlation spectrum measured with 53-ms CORD mixing detects all intramolecular cross peaks of chitin and four types of glucans. Abbreviations are used for resonance assignment and different polysaccharide signals are color coded. **b** ¹³C CP J-INADEQUATE spectrum resolves the ¹³C through-bond connectivity for each polysaccharide. **c** Identified polysaccharides and representative chemical shifts. All spectra were measured on an 800 MHz solid-state NMR spectrometer

Interestingly, the 2D ¹³C–¹³C correlation spectrum of the alkali-insoluble portion of cell walls from a related but non-pathogenic fungus, *Aspergillus niger*, also shows a comparable pattern to that of intact *A. fumigatus*, with signals primarily from chitin, β-1,3-glucan and α-1,3-glucan (Supplementary Fig. 2). Linkage analysis confirmed that more than 80% of neutral sugars are 3-Glc*p* residues (Table 1). Therefore, the dominance of chitin, β-1,3-glucan, and α-1,3-glucan in the stiff core of cell walls is a conserved feature in *Aspergillus* species.

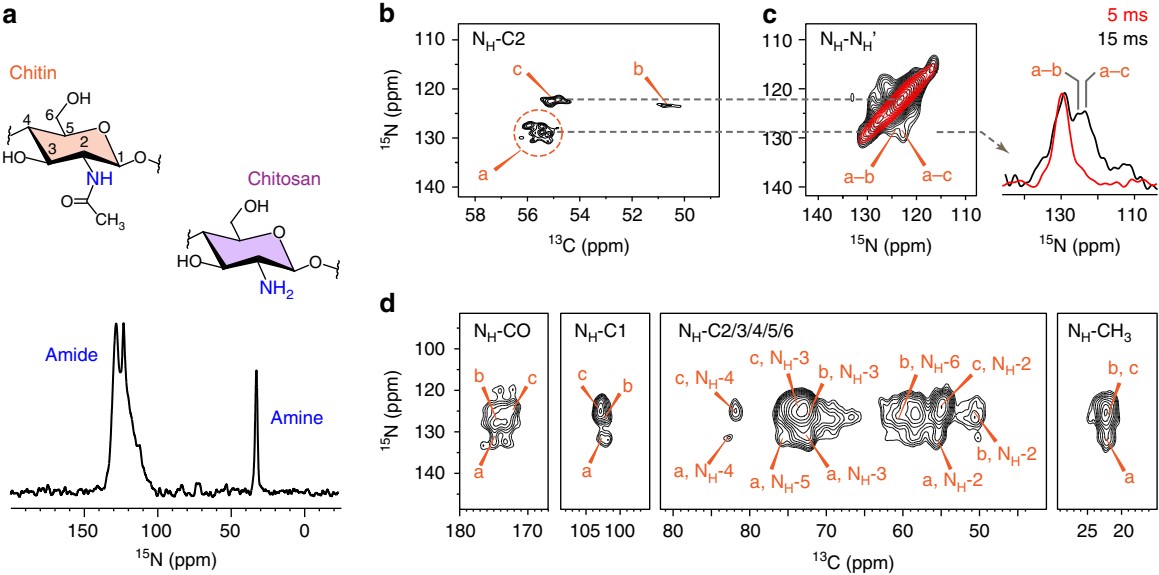

**Fig. 2** Chitin is structurally polymorphic in intact *A. fumigatus* cell walls. **a** 1D $^{15}N$ spectrum resolved multiple amide and amine signals. **b** High-resolution $^{15}N$–$^{13}C$ correlations spectra resolved three major types of chitins. **c** DNP-assisted $^{15}N$–$^{15}N$ PAR spectra measured using 5 ms (red) and 15 ms (black) mixing times revealed extensive cross peaks between different chitin allomorphs. **d** DNP-assisted $^{15}N$–$^{13}C$ correlation spectra measured using 3-mg *A. fumigatus*

**Fungal polysaccharides are highly polymorphic in structure**. The decent $^{13}C$ resolution of the *A. fumigatus* sample allows the unambiguous assignment of seven types of polysaccharides, including chitin, α-1,3-glucan, three types of β-glucans, mannan, and arabinan. In total 23 allomorphs have been identified for these molecules. 1D $^{15}N$ spectrum revealed multiple major types of nitrogenated polysaccharides, with two peaks of chitin amides at 123 and 128 ppm and a sharp amine peak at 33 ppm from other amino sugars such as chitosan. Chitosan is an enzymatical derivative of chitin and the degree of deacetylation (DD)[42] is 10% as determined using the well-resolved $^{15}N$ signals (Fig. 2a).

Chitin is found to be the most polymorphic molecule in fungal cell walls. 2D $^{15}N$–$^{13}C$ correlation spectrum resolves three major forms of chitin (Fig. 2b), and just for the type-a chitin, we can resolve at least six C2-$N_H$ cross peaks, each representing a different sub-form. With another three minor types identified using DNP, we have discovered 11 types of chitins (Supplementary Table 2). This level of structural polymorphism is beyond our current knowledge obtained from X-ray and NMR studies on model chitins, and the 11 types of signals could not be explained using the known ways of packing: the antiparallel α-chitin, parallel β-chitin, and the mixed γ-chitin[43,44]. Signals from the known structures, both the α and β allomorphs[45], can be found within the signals of type-a chitin, with a per-carbon chemical shift difference as low as 0.3 ppm. In contrast, types b and c do not correlate with any known structures, and the chemical shift difference increased to at least 0.7 ppm and 1.2 ppm, respectively. Thus, chitin b and c belong to two structures that have never been reported before. This unexpected level of structural polymorphism is potentially caused by the sophisticated pattern of hydrogen-bonding through the N–H and C=O groups in chitin when multiple chains are put together outside the plasma membrane after the biosynthesis.

It is striking that the three major forms of chitin identified in this study are thoroughly mixed on the subnanometer scale in the intact *A. fumigatus* cell wall. This is revealed by the strong off-diagonal cross peaks between types a and b and between types a and c observed in the 15-ms $^{15}N$–$^{15}N$ proton-assisted recoupling (PAR) spectrum (Fig. 2c)[46,47]. For these interactions to happen in

the microfibrils, chitin allomorphs should coexist as tightly packed chains in the fibrillar cross-section rather than as separated domains associated longitudinally along the fibril. This organization of chitin bears a resemblance to the assembly of glucan chains in plant cellulose, in which seven types of glucan chains are found to coexist in the cross-section of a single microfibril[48]. It should be noted that the presence of amide, methyl, and carbonyl groups substantially facilitates the resonance assignment of chitin allomorphs (Fig. 2d). This unique chemical structure further serves as the basis for spectral editing to determine the chitin–glucan packing (vide infra).

**MAS-DNP reveals a tight packing of chitin and α-1,3-glucan.** It has been a long-standing question of how cell wall biomolecules interact to form the polymer network. To address this question, we measured a 15-ms $^{13}C$–$^{13}C$ PAR spectrum and discovered 23 long range (5–10 Å) intermolecular cross peaks (Fig. 3a and Supplementary Fig. 3). Most of these cross peaks originate from intermolecular interactions between chitin (Ch) and α-1,3-glucans (A), for instance, between type-b α-1,3-glucan carbon 1 and type-b chitin carbon 4 (A1[b]–Ch4[b]) and between type-a/c α-1,3-glucan carbon 3 and type-a chitin carbon 4 (A3[a,c]–Ch4[a]). In addition, several cross peaks are also found between the chitin-α-1,3-glucan complex and β-glucans. Noteworthy examples include the α-1,3-glucan carbon 1 to β-1,6-glucan carbon 3 (A1–H3) cross peak at (101, 79) ppm and α-1,3-glucan carbon 3 to β-1,3-glucan carbon 4 (A3–B4) cross peak at (85, 73) ppm. Therefore, despite the close packing of chitin and α-1,3-glucan, these two molecules are still crosslinked by β-glucans.

To concurrently improve the sensitivity and resolution, we combined the DNP technique[49,50] with spectral-editing methods and successfully identified another 35 long-range interactions. The magic-angle spinning DNP (MAS-DNP) enhances the NMR sensitivity by tens to hundreds of fold by transferring the polarization from the electrons in radicals to NMR-active nuclei in biomolecules under microwave (μW) irradiation[29]. With an optimized protocol ensuring homogeneous mixing of radicals with cell wall biomolecules, a sensitivity enhancement ($\varepsilon_{on/off}$) of

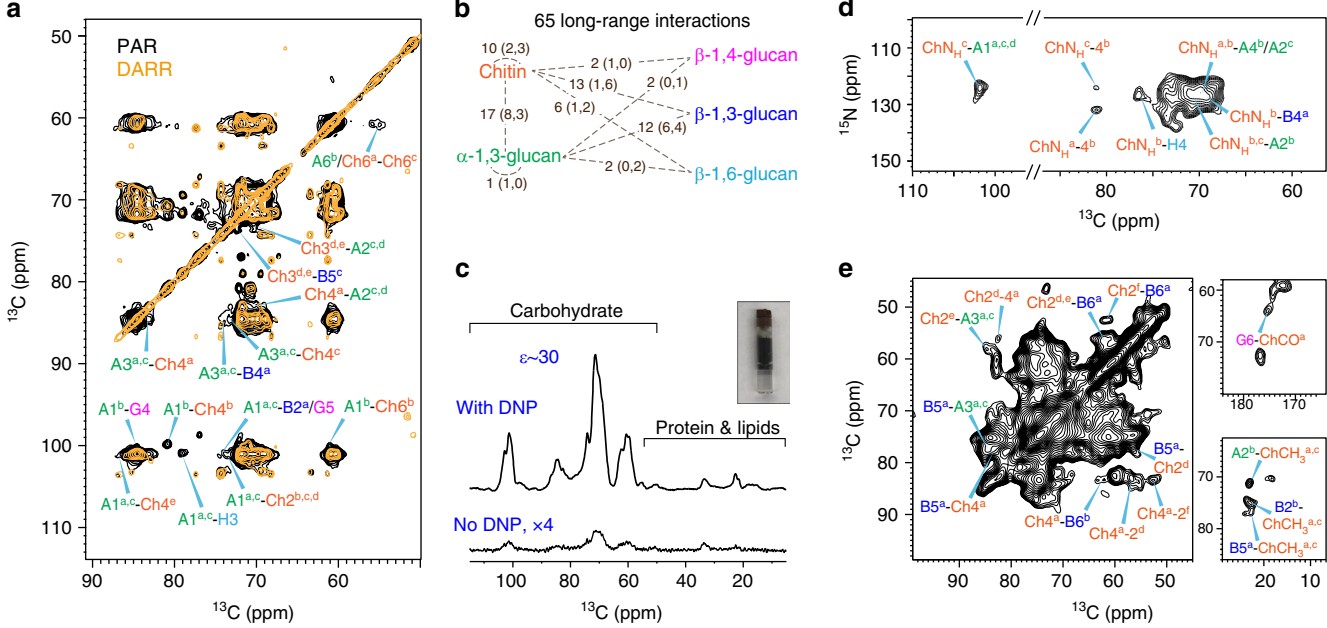

**Fig. 3** MAS-DNP solid-state NMR reveals the tight packing of chitin and α-1,3-glucan. **a** 15 ms $^{13}$C–$^{13}$C PAR spectrum (black) reports many intermolecular cross peaks, mainly between chitin and α-1,3-glucans. A 100-ms DARR spectrum (orange) that primarily detects intramolecular correlations is overlaid for comparison. **b** Summary of 65 long-range restraints. For each category, the number of all cross peaks and the number of strong and intermediate restraints (in parenthesis) are listed. **c** Sensitivity enhancement $\varepsilon_{on/off}$ of 30-fold was obtained for *A. fumigatus*. A picture of a DNP sample is also included. **d** DNP-assisted intermolecular-only $^{15}$N-$^{13}$C correlation spectrum unambiguously detected several chitin–glucan cross peaks. **e** DNP-assisted chitin-edited spectrum only shows signals from chitin itself or the glucans that are spatially proximal. The DNP experiments were conducted on a 600 MHz/395 GHz spectrometer

30-fold has been achieved using the intact cells of $^{13}$C, $^{15}$N-labeled *A. fumigatus* on a 600 MHz/395 GHz DNP spectrometer (Fig. 3c). The feasible sensitivity not only facilitates the detection of long-range cross peaks with weak intensities but also allows us to employ spectral-editing methods to alleviate the signal overlapping issue in intact cells. Briefly, the $^{15}$N magnetization of chitin amide is first selected through a $^{15}$N–$^{13}$C dipolar filter[51,52] and then transferred to spatially proximal glucans via a proton-driven spin diffusion (PDSD) mixing period (Supplementary Fig. 4). By subtracting two parent $^{15}$N–$^{13}$C correlation spectra measured with short (100 ms) and long (1 s) mixing times, we can eliminate all intramolecular signals[53] (Supplementary Fig. 5). The resulting spectrum only contains long-range intermolecular cross peaks that are structurally important (Fig. 3d). Using this method, we have identified seven additional cross peaks between different chitin allomorphs, between the chitin and α-glucan, such as the chitin nitrogen to α-1,3-glucan carbon 1 (ChN$_H$-A1) cross peak, and between chitin amides and β-glucan carbons such as the ChN$_H$–H4 and ChN$_H$–B4 cross peaks. The same strategy is extended to measure chitin-edited 2D $^{13}$C–$^{13}$C correlation spectrum, which enables the identification of another 25 long-range cross peaks, for instance, between chitin carbonyl/methyl groups to β-1,3- or β-1,6-glucans (Fig. 3e).

Overall, these experiments have generated 65 long-range cross peaks, among which 54 are intermolecular interactions and 11 are inter-allomorph cross peaks within chitin or α-1,3-glucans (Fig. 3b). The cross peaks are further categorized into 20 strong, 21 medium, and 24 weak restraints according to the relative intensity and the experimental methods (Supplementary Table 3). A total of 8 out of the 17 cross peaks between chitin and α-1,3-glucan are strong, constituting 40% of all strong restraints, supporting the proposed complex formed by tightly packed chitin and α-1,3-glucans. These two molecules further exhibit 25

through-space cross peaks to β-1,3-glucans, among which only 7 are strong restraints, therefore, the β-1,3-glucan, assisted by β-1,4- and β-1,6-glucans, may serve as tethers between multiple chitin-α-1,3-glucans segments. This finding is further supported by the high hydration level and mobility of β-1,3-glucan (vide infra).

**Chitin and α-1,3-glucan form a hydrophobic core**. To investigate carbohydrate–water interactions, we conducted the water-edited 2D $^{13}$C–$^{13}$C correlation experiment[54–56]. This experiment relies on a $^1$H-T$_2$ relaxation filter to eliminate all original polysaccharide signals and then transfers the water $^1$H magnetization to carbohydrates so that only carbohydrates with bound water can be detected. The water-edited signals are compared with the equilibrium intensities of a control spectrum (Fig. 4a, b), and the intensity ratios tell the polymer hydration in a site-specific manner. Among all the complex carbohydrates, α-1,3-glucan and chitin are most hydrophobic and have the lowest water-transferred intensity. For instance, all α-1,3-glucan cross peaks, including A4-1 (carbons 4 to carbon 1), A4-3, A4-2, and A4-6, show substantial dephasing in the 2D water-edited spectrum (Fig. 4a) and its 69.5-ppm $^{13}$C cross-section (Fig. 4b). The residual intensities are below 40% for chitin and α-1,3-glucan but are higher than 60% for all the well-hydrated β-glucans (Fig. 4c and Supplementary Table 4). The hydration data dovetail nicely with the long-range correlation results, collectively indicating that chitin and α-1,3-glucan tightly pack to form the hydrophobic domains that are surrounded by a well-hydrated matrix formed of β-glucans. The formation of this dehydrated domain might be facilitated by the assembly of chitin microfibrils, with α-1,3-glucans occasionally deposited on the fibrillar surface or between multiple chains or several microfibrils to further reduce the water accessibility.

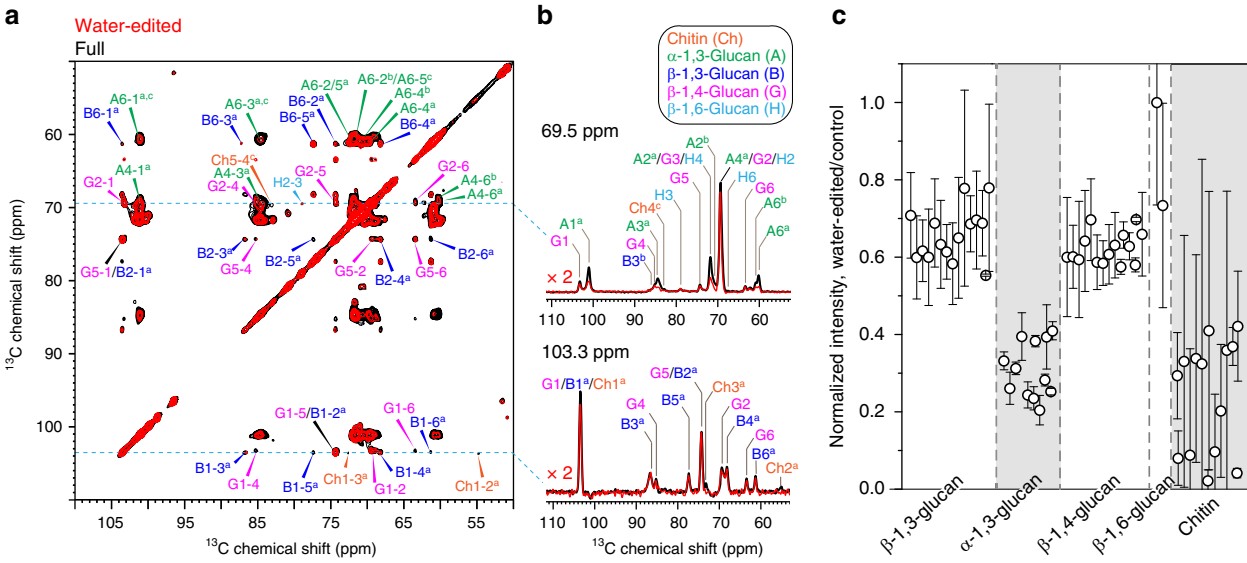

**Fig. 4** Chitin and α-1,3-glucans form the hydrophobic core of *A. fumigatus* cell walls. **a** Overlay of 2D water-edited (red) and control (black) $^{13}C$–$^{13}C$ correlation spectra and **b** 1D $^{13}C$ cross sections. **c** Relative intensities of different polysaccharides in water-edited spectra. The water-edited spectrum preferentially detects well-hydrated molecules. Error bars indicate standard deviations propagated from NMR signal-to-noise ratios. Chitin and α-1,3-glucans have the lowest intensities in water-edited spectra (shaded)

**Rigid chitin and α-glucan are enclosed by mobile β-glucans.** To systematically examine the dynamics of cell wall biomolecules, we measured NMR relaxations and a series of 1D $^{13}C$ spectra that select components with different mobilities. The 1D $^{13}C$ spectrum measured using $^{13}C$ direct polarization (DP) and a long recycle delay of 35 s provides quantitative detection of all cell wall components including polysaccharides, proteins, lipids and small molecules (Fig. 5a). This quantitative spectrum differs substantially from the spectrum that favorably detects mobile molecules through the combined use of $^{13}C$ DP and short recycle delays of 2 s. The difference between these two DP spectra is comparable to the $^{13}C$ CP spectrum that favors the rigid components with stronger $^{1}H$–$^{13}C$ dipolar couplings (Fig. 5b, c). The major peaks in this difference spectrum are from chitin and α-1,3-glucan, unveiling the stiffness of these two molecules. At the same time, signals from relatively mobile components, such as proteins, lipids, β-glucans, and small molecules, are substantially suppressed (Fig. 5b).

When compared with the quantitative detection, α-1,3-glucans have the highest intensity in the CP spectrum selecting stiff polymers, doubling that of quantitative DP, but the lowest signals in the 2-s DP spectrum that favors mobile molecules (Fig. 5d). Accordingly, α-1,3-glucan is the most rigid polysaccharides in *A. fumigatus*. Two subtypes, a and b, of chitin are found to be the second most rigid molecules, with 50% higher signals in CP but 0–50% less intensity in 2-s DP spectrum than the quantitative spectrum. All the β-glucans, together with another two types of chitins, fully retain their signals in 2-s DP spectrum, thus are dynamic (Fig. 5d and Supplementary Table 5).

The molecular motional rates are quantified by measurements of $^{13}C$-$T_1$ relaxation that probes local reorientation and $^{1}H$-$T_{1\rho}$ for larger-scale motions, such as cooperative movement of multiple sugar rings. Cell wall polysaccharides exhibit distinct $^{1}H$-$T_{1\rho}$ relaxation times, signifying a highly heterogeneous profile of dynamics (Supplementary Fig. 6a and Supplementary Table 6). The average $^{1}H$-$T_{1\rho}$ is 4.5 ms, 2.8 ms, and 1.1 ms for α-1,3-glucans, chitin, and β-glucan, respectively (Fig. 5e). Consistently, α-1,3-glucans also have the slowest $^{13}C$-$T_1$ relaxation among all

types of glucans (Supplementary Fig. 6b and Supplementary Table 7). Therefore, α-1,3-glucan is the most rigid molecule, followed by chitins and then β-glucans.

Interestingly, chitin has the largest variance in relaxation times (Fig. 5e), which, together with the allomorph-specific 1D $^{13}C$ intensity, suggests a functional-relevant structural polymorphism: a subgroup of chitin allomorphs are responsible for forming rigid microfibrils whereas the remainders retain considerable disorder due to unfavored conformations or unstable hydrogen-bonding patterns. Type-a chitin has similar chemical shifts as the α and β chitins, and these allomorphs clearly bear high rigidity in the intact cell walls. It should be noted that for most α-1,3-glucan peaks the satisfactory fit of $^{1}H$-$T_{1\rho}$ data can only be achieved using double exponential equations. This suggests a two-domain distribution: 70–90% of α-1,3-glucans interact with chitins to form a stiff and hydrophobic scaffold conferring rigidity to the cell wall (the long $^{1}H$-$T_{1\rho}$ component of 3.8–5.0 ms) while the other 10–30% have very short $^{1}H$-$T_{1\rho}$ relaxation of less than a millisecond due to the interactions with the mobile β-glucans (Fig. 5e and Supplementary Fig. 6a). Particularly, β-1,3-glucan is the major binding target of α-1,3-glucans as they have 12 intermolecular cross peaks, among which half are strong interactions (Fig. 3b).

**Glycoproteins and α-1,3-glucans form a highly mobile shell.** While the previous 2D experiments were CP-based and primarily focused on the relatively rigid polysaccharides that are mechanically important, a $^{13}C$ DP J-INADEQUATE spectrum with a short recycle delay of 2-s is also measured to highlight the mobile domain of cell walls (Fig. 6a). The primary signals have been attributed to proteins and some polysaccharides. The very sharp $^{13}C$ linewidths of 0.3–0.5 ppm confirmed the rapid motional averaging of these molecules. Unambiguous signals of mannan and arabinan are observed (Fig. 6b). Since mannan is a major component of fungal glycoproteins purposely forming an outmost layer of fungal cell walls[1,15], our results reveal that this outer shell is highly dynamic and spatially separated from the relatively rigid inner domain of chitin and glucans. Despite the

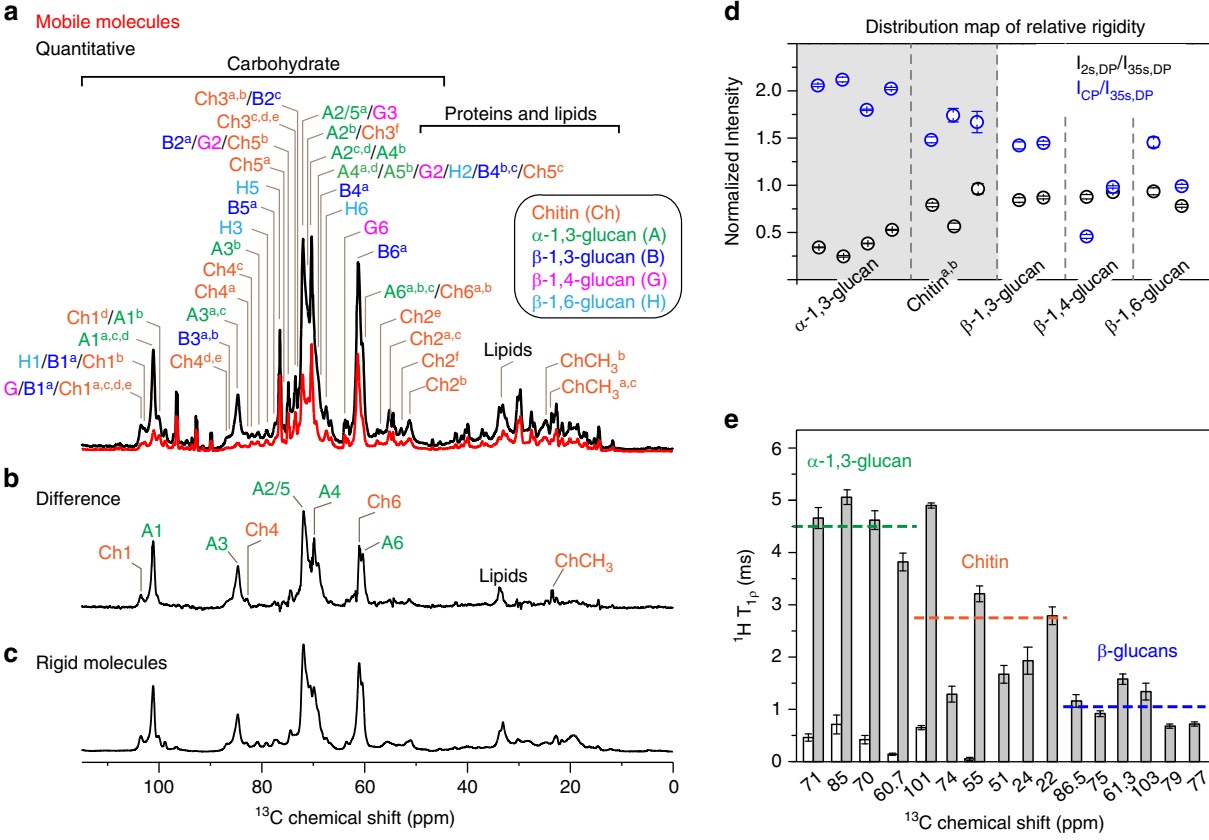

**Fig. 5** *A. fumigatus* cell walls are dynamically heterogeneous. **a** 1D $^{13}$C DP spectra of intact *A. fumigatus* cells with quantitative detection of all molecules (black) or selective detection of the mobile components (red). Abbreviations of carbohydrate names, carbon numbers, and subtypes (superscripts) are included in the assignment. **b** Difference spectrum obtained by subtraction of the two 1D DP spectra. **c** $^{13}$C CP spectrum that favors rigid molecules. **d** Peak intensity ratios between different DP spectra and the CP spectrum show that α-1,3-glucan and a subset of chitins are relatively rigid. Error bars are standard deviations propagated from NMR signal-to-noise ratios. **e** $^1$H-T$_{1ρ}$ relaxation times. The open and filled bars represent the short and long-components of $^1$H-T$_{1ρ}$ relaxation. Error bars are standard deviations of the fit parameters. Dashlines indicate the average value of the longer $^1$H-T$_{1ρ}$ component for each type of polysaccharides

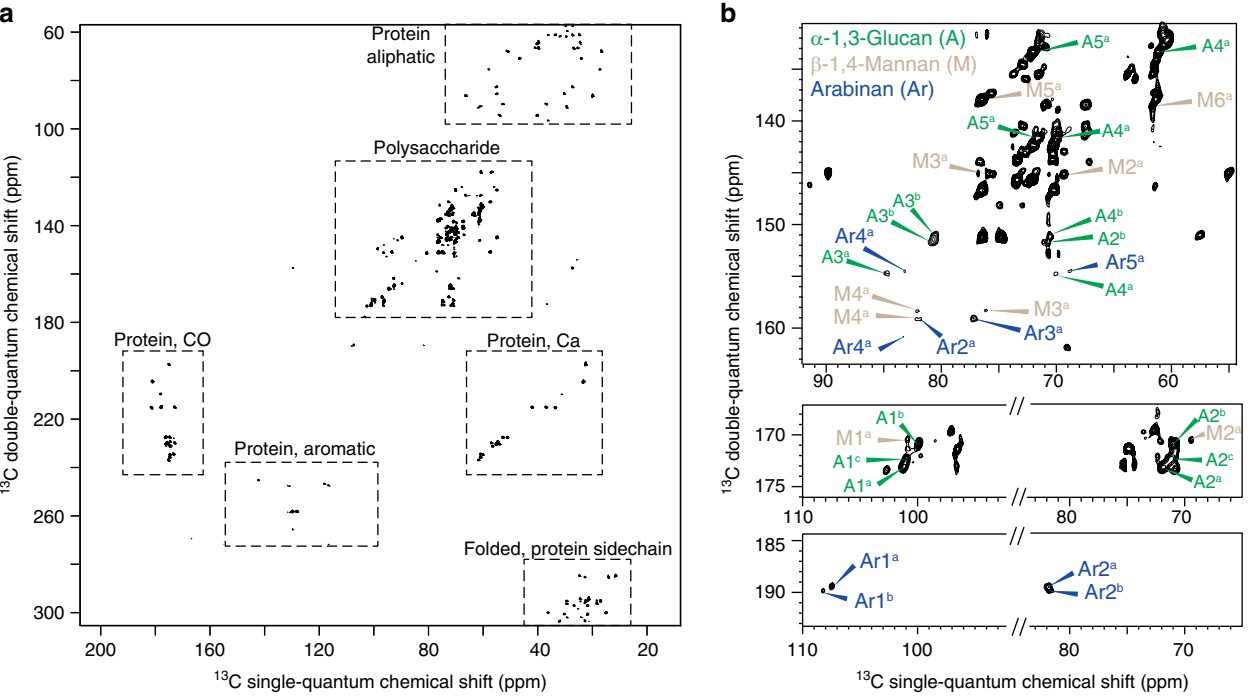

**Fig. 6** Glycoprotein and α-1,3-glucan form a highly mobile shell. **a** 2D $^{13}$C DP J-INADEQUATE spectrum selects only the very mobile molecules of proteins and polysaccharides. **b** The polysaccharide region reveals the presence of β-1,4-mannan, arabinan and α-1,3-glucan

fact that most α-1,3-glucans participate in the formation of stiff and hydrophobic cores, a considerable amount of α-1,3-glucan signals remain in this dynamic domain (Fig. 6b), which supports the hypothesis that α-1,3-glucans may form the outmost layer of cell walls to block the immune recognition of β-glucan receptors in the host cells[11,57]. Thus, α-1,3-glucan is bi-functional: supporting cell walls through the formation of hydrophobic scaffolds and potentially increasing fungal virulence by disabling the host detection of invading microbes.

In addition to this mobile domain, we have also identified a rigid component of proteins (Supplementary Fig. 7a and Supplementary Table 8). These proteins are as hydrophobic as the fatty acid chains of the membrane and are more hydrophobic than any polysaccharides (Supplementary Fig. 7b and Supplementary Table 9). These rigid proteins are mainly membrane proteins but may also exist in the hydrophobin rodlet protein layer, a hydrophobic coating preventing the immune recognition[58,59].

## Discussion

This study presents a high-resolution and non-destructive method for determining the architecture of fungal cell walls. Solid-state NMR and MAS-DNP results of 65 intermolecular and interallomorph interactions, site-specific hydration, and molecular mobility steadily indicate a two-domain distribution of molecules: glucans and chitins form a relatively rigid and inner portion of cell walls, while mannoproteins and α-1,3-glucan form the extremely mobile outer shell. In the inner domain, α-1,3-glucan and chitin are tightly packed as the most rigid and hydrophobic cores that are embedded in a well-hydrated and relatively mobile matrix formed by β-1,3-, β-1,4-, and β-1,6-glucans (Fig. 7).

Compared with previous biochemical analyses, this NMR-derived model has both consistency and revision. For decades, we have been solubilizing individual components of the cell wall using chemical or enzymatical treatment, for example, alkali solubilization, and then determine the composition and covalent linkages of the extracted portions[17,18,60–64]. A conserved skeleton of branched β-1,3 and 1,6-glucans with β-1,4-linkages to

chitins is found in the alkali-insoluble component of most fungi, while the other molecules vary substantially in mold and yeast[16,60]. Mannoproteins are found in the cell wall surface and connected to the cell wall via either non-covalent connections or covalent linkages to β-1,6-glucans[65,66]. These biochemical data and our ssNMR analysis dovetail well considering the structural roles of chitin and β-glucans, as well as the outer layer of proteins.

The current model also shifts the prevailing paradigm of fungal cell wall in three aspects. First, we have identified the molecules that determine cell wall rigidity. As the most abundant building unit of the fungal cell wall, β-glucans, especially the multi-branched polymers of β-1,3/-1,6-glucan, have long been proposed to form the rigid network[15,16]. However, the high level of hydration and the intermediate mobility of β-glucans we found have clearly excluded this structural role. Instead, we found that the rigid scaffold are formed by chitin and α-1,3-glucan. The structural role of α-1,3-glucan is unexpected since it is usually defined as the major alkali-soluble polysaccharide in previous studies[16]. Second, the current model reveals the bridging role of β-1,3-glucans to a great detail. At present, all β-glucans are indistinguishable in dynamics and hydration, but among the three types of β-glucans, β-1,3-glucan absolutely has more pronounced interactions with other molecules, in particular, with α-1,3-glucan. This can be seen from the 12 intermolecular cross peaks between β-1,3-glucan and α-1,3-glucan in which half are strong restraints (Fig. 3b). β-1,3-glucan and chitin also have a large number of cross peaks, 13 in total, most of which are only weak or intermediate in strength. Thus, chitin may serve as a secondary anchor, following α-1,3-glucan, for β-1,3-glucan to link the rigid and mobile domains (Fig. 7). Third, the dual functions of α-1,3-glucan have been emphasized. The high rigidity of α-1,3-glucan has not been expected and we propose that the stiffening by covalent or physical interactions with β-1,3-glucans and chitin grant α-1,3-glucan with the capability of performing structural roles. Linkages between α-1,3- and α-1,4-glucose units have been reported previously[61], but it is unclear whether covalent interactions exist between α-1,3-glucans and chitin or β-1,3-glucans. Interestingly, the occurrence of

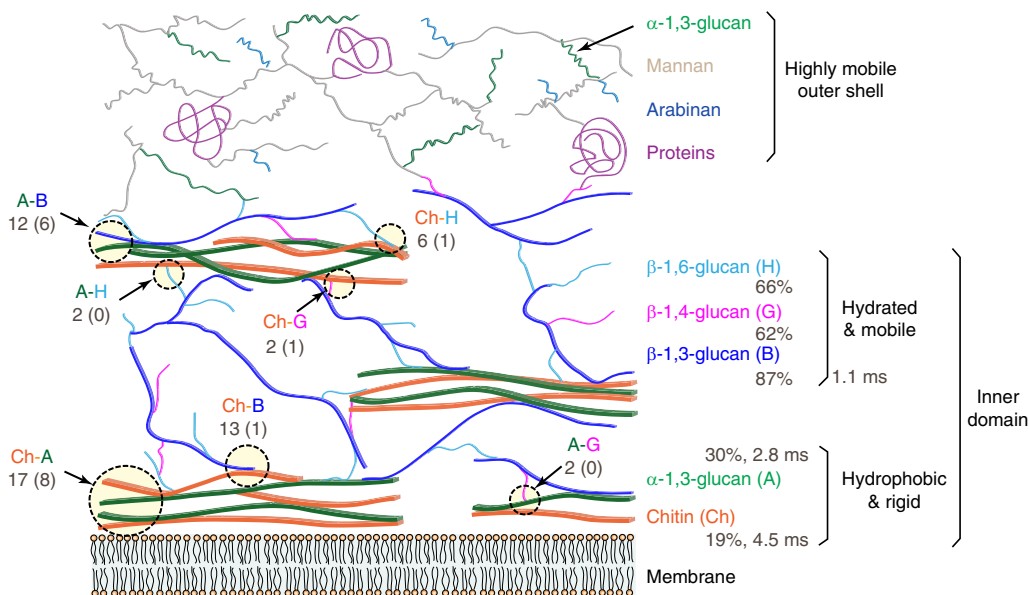

**Fig. 7** Illustrative model of the supramolecular architecture of *A. fumigatus* cell walls. Dashline circles highlight the intermolecular interactions with the total numbers of NMR restraints indicated. The number of strong restraints is in parenthesis. The average hydration levels (percentage values) and the average $^1$H-T$_{1\rho}$ relaxation times (in millisecond) of each polymer are also labeled

α-1,3-glucans in two distinct domains, the outer surface and the inner rigid cores, demonstrated the structural and functional versatility of this molecule.

*A. fumigatus* produces a low amount of chitosan, a cationic molecule with various biomedical and industrial applications due to its antimicrobial, anti-tumor, wound-healing, and cholesterol-lowering properties[67]. Previously, structural roles have been assumed for chitosan since the chitosan-deficient strains of the pathogenic fungus *Cryptococcus neoformans* have compromised cell wall integrity in vitro, resulting in attenuated virulence[68]. However, in *A. fumigatus*, chitosan accounts for <1% of all carbohydrates (Supplementary Table 1) and remains mobile as revealed by the sharp $^{15}N$ linewidth of 1.5 ppm (Fig. 2a). Therefore, chitosan contributes negligibly to cell wall rigidity in *A. fumigatus*.

The current study also provides insights into the polysaccharide scaffold for pigment deposition. The insoluble pigment of melanin increases cell hydrophobicity and reduces cell wall porosity, which was proposed to be the cause of drug resistance in many fungi[69–72]. Recent ssNMR studies of melanin "ghosts" indicated that melanin may be integrated into the cell wall via association with the polysaccharides in *C. neoformans*[27,28]. However, the scaffold that holds the pigment in place was unclear. Chitin has been proposed to be the supporting polysaccharide, but our results revealed a stiff and hydrophobic frame of α-1,3-glucans and chitins that could accommodate the aromatic assembly of pigments, which, in turn, may provide an explanation for the high hydrophobicity of these two polysaccharides (Fig. 4c).

Notably, the fungal wall is significantly more dynamic than its counterparts in plants. The narrow $^{13}C$ linewidth of fungal polysaccharides is comparable to that of the matrix polysaccharides in the fast-growing primary plant cell walls, but is apparently narrower than that of rigid cellulose microfibrils[48]. Upon maturation, plant cell walls are further rigidified and dehydrated by the deposition of lignin and the coalescence of cellulose microfibrils[73]. Therefore, despite the presence of a relatively rigid scaffold formed by chitin fibrils and α-1,3-glucans, the polymer network in fungi still retains considerable plasticity, which allows the fungal cell wall to reshape its molecular architecture to survive through different external stress and to fulfill diverse functions. The lack of need for vertical growth and the limited size of microbes may also explain the high mobility of fungal cell walls. In addition, the fungal cell wall also has a substantially larger number of intermolecular cross peaks than the primary plant cell walls. This may be caused by the extensive covalent cross-linking of glucans and chitin in fungi, while plants principally rely on non-covalent interactions, such as van der Waals forces and electrostatic interactions, as well as the entanglement and entrapment of polymers. Given the high mobility of fungal biomolecules, it may be crucial to have chemical linkages instead of physical contacts as the primary interactions so that this dynamic structure could remain intact under external and internal stress.

Although this study builds a frame for the fungal cell wall structure, in particular, for the complex carbohydrates, more structural details await systematic investigations. Emerging questions include the correlation of linkage patterns with structural or mechanical functions for β-glucans and the developmental changes of cell wall architecture, for example, during conidial outgrowth. A detailed and dynamic understanding of the supramolecular assembly of fungal cell walls and future studies on drug response will substantially facilitate the design of better antifungal compounds that inhibit a broader spectrum of invasive fungal infections with minimal or no side toxicity.

## Methods

**Preparation of fungal materials**. Uniformly $^{13}C,^{15}N$-labeled mycelium was prepared by IsoLife (Wageningen, The Netherlands) using the following protocol. *Aspergillus fumigatus* cultures (strain RL 578, a wild strain obtained from compost) were grown on a $^{13}C/^{15}N$ liquid medium (a modified Czapek-Dox medium) under still cultivation at 30 °C for 14 days in the dark on 50-mL medium in 250 mL capacity Erlenmeyer flasks. At the end of the cultivation the mycelium was flash-frozen in liquid nitrogen and stored at −80 °C. The resulting materials were further dialyzed for six times over 3 days to remove the majority of small molecules from the isotope-labeled media and to reduce the ion concentration. For solid-state NMR experiments, 30 and 100 mg of this whole-cell material was packed into 3.2-mm and 4-mm magic-angle spinning (MAS) rotors, respectively. Another 5 mg was proceeded in the DNP matrix for DNP experiments and 3 mg was finally transferred into a 3.2-mm sapphire rotor. The hydration level is 90% for the initial culture and has been decreased to around 50% in the NMR samples by multiple times of compression.

To verify the composition of the rigid portion of fungal cell walls and compare it with the whole-cell, the alkali-insoluble polysaccharides were extracted by IsoLife from *A. niger* mycelium by deproteinization with 2% w/v sodium hydroxide solution (30:1 v/w, 90 °C, 2 h), separation of alkali-insoluble fraction by centrifugation (4000 × *g*, 15 min), extraction of chitosan under reflux (10% v/v acetic acid, 40:1 v/w, 60 °C, 6 h), separation of crude cell walls by centrifugation (4000 × *g*, 15 min). The crude was washed with water, ethanol and acetone and then air-dried at 20 °C. A total of 70 mg of the extracted polysaccharides was packed in a 4-mm rotor for solid-state NMR experiments.

**Glycosyl composition analysis**. Glycosyl composition analysis of neutral sugars constituting the cell wall was achieved after the dispersion of 600 μg dry samples in 0.5 mL 2 M TFA (v/v) in sealed reaction tubes, by 20 min sonication in an ultra-sound water bath at RT, followed by 2 h of hydrolysis at 121 °C, overnight reduction with $NaBD_4$, and 1 h acetylation with acetic anhydride and pyridine at 80 °C. Inositol was used as an internal standard. The glycosyl constituents were assigned based on the retention time of the derivatives of original sugar standards and unique EI-MS fragments of $^{13}C$ alditol acetate derivatives identified in the fungal samples. We used the same GLC-MS equipment and temperature program as for linkage analysis.

**Glycosyl linkage analysis**. The glycosyl linkage of $^{13}C,^{15}N$ uniformly labeled polysaccharides was obtained by GC-MS analysis of partially methylated alditol acetates (PMMA)[74] after 2 h hydrolysis with 2 M (v/v) TFA at 121 °C, overnight reduction with $NaBD_4$ and acetylation with acetic anhydride and pyridine. The inositol was used as an internal standard. GLC-MS analysis was performed on an HP-5890 GC interfaced to a mass selective detector 5970 MSD using a Supelco SP2330 capillary column (30 × 0.25 mm ID, Supelco) with the following temperature program: 60 °C for 1 min, then ramp to 170 °C at 27.5 °C/min, and to 235 °C at 4 °C/min with 2 min hold and finally at to 240 °C at 3 °C/min with 12 min hold.

Because the organism was grown in media supplemented with $^{13}C$ and $^{15}N$ as the sole carbon and nitrogen source, respectively, one could expect these isotopes to be incorporated in the cell wall polysaccharides. Consequently, GC-MS analysis of partially methylated alditol acetates (PMAA) generated a set of unique EI-MS diagnostic fragments that differed from fragments predicted for classical PMAAs (Supplementary Fig. 1). It should be noted that the complex glucan samples solubilize better during the linkage analysis (DMSO solvent and permethylation step prior to TFA hydrolysis) than in the classical alditol acetate method of compositional analysis (only TFA hydrolysis). Thus, we have a better detection of glucans in the PMMA analysis.

**Solid-state NMR experiments**. Solid-state NMR experiments were conducted on a Bruker Avance 800 MHz (18.8 Tesla) spectrometer and a 400 MHz (9.4 Tesla) Bruker Avance spectrometer using 3.2-mm and 4-mm MAS HCN probes, respectively. Most experiments except those with MAS-DNP were collected under 10–13.5 kHz MAS at 290 K. $^{13}C$ chemical shifts were externally referenced to the adamantane $CH_2$ signal at 38.48 ppm on the TMS scale. $^{15}N$ chemical shifts were referenced to the liquid ammonia scale either externally through the methionine amide resonance (127.88 ppm) of the model peptide *N*-formyl-Met-Leu-Phe-OH[75] or using the ratio of the gyromagnetic ratios of $^{15}N$ and $^{13}C$. Typical radio-frequency field strengths, unless specifically mentioned, were 80–100 kHz for $^1H$ decoupling, 62.5 kHz for $^1H$ CP and hard pulses, and 50–62.5 kHz for $^{13}C$ and 41 kHz for $^{15}N$.

To assign the $^{13}C$ and $^{15}N$ resonances of cell wall biomolecules, five types of experiments were measured: (1) 2D refocused $^{13}C$ J-INADEQUATE[40,41] spectra with DP and short recycle delays of 2 s for selective detection of the very mobile phase in the outer layer of cell walls; (2) $^{13}C$ CP J-INADEQUATE for the detection of rigid molecules located inside the cell wall; (3) $^{13}C–^{13}C$ RFDR[76] for the detection of one-bond cross peaks; (4) 2D $^{13}C–^{13}C$ spectra using 53 ms CORD[38,39] or 50 ms DARR for the detection of intramolecular cross peaks; (5) 2D $^{15}N–^{13}C$ N(CA)CX heteronuclear correlation spectra[77] to select the amide signals of chitins. The N(CA)CX is measured using a 0.6 ms $^1H–^{15}N$ CP contact time, a 5 ms $^{15}N–^{13}C$ CP

and a 100 ms $^{13}$C–$^{13}$C DARR mixing period. The $^{15}$N and $^{13}$C spin-lock field strengths for the $^{15}$N–$^{13}$C CP were 20 kHz and 33 kHz, respectively. The $^{13}$C and $^{15}$N carrier frequencies were 55 ppm and 70 ppm, respectively. A strong $^{1}$H decoupling of 100 kHz is applied during the $^{15}$N–$^{13}$C CP.

To determine the spatial proximities of biomolecules in intact cell walls, we measured 15-ms $^{13}$C–$^{13}$C PAR[78] using $^{13}$C field strengths of 53 kHz and $^{1}$H field strengths of 50 kHz. A total of 23 intermolecular cross peaks have been identified in the PAR spectra, which, in combination with 7 long-range cross peaks in 3-s PDSD and 35 long-range cross peaks in DNP experiments, restrain the spatial packing of molecules in intact cell walls.

To site specifically determine the water accessibilities of different polysaccharides, we measured water-edited 2D $^{13}$C–$^{13}$C correlation spectra[54,55,79]. This experiment initiates with $^{1}$H excitation followed by a $^{1}$H-$T_2$ filter of 0.88 ms × 2 that eliminates 97% of polysaccharide signals but retains 80% of water magnetization, a 4-ms $^{1}$H mixing period for water-to-polysaccharide transfer and a 1 ms $^{1}$H–$^{13}$C CP for $^{13}$C detection. A 50-ms DARR mixing period is used for both the water-edited spectrum and the control 2D spectrum showing full intensity. The intensity ratio between the water-edited spectrum and the control spectrum is quantified, which is further normalized by that of the C2–C3 cross peak of β-1,6-glucan, the highest value among all the peaks, to reflect the relative degree of hydration.

To examine the dynamics of polysaccharides, we measured a series of 1D $^{13}$C spectra with different methods for the creation of initial magnetization. 1D $^{13}$C direct polarization (DP) spectra were measured using a 35 s recycle delay to obtain the quantitative signals and a 2 s recycle delay to selectively detect the dynamic components. The difference spectrum was obtained by subtracting the 2 s DP spectrum from the 35 s DP spectrum, without scaling. 1D $^{13}$C CP spectrum that preferentially detects the rigid components was also conducted using 1-ms contact time. Furthermore, we measured both $^{13}$C spin-lattice ($T_1$) relaxation and $^{1}$H rotating-frame spin-lattice relaxation ($T_{1\rho}$) at 298 K under 10 kHz MAS on a Bruker Avance 400-MHz spectrometer. The spin-lock field was 62.5 kHz for the measurement of $^{1}$H-$T_{1\rho}$. The relaxation data were fit using either a double or single exponential decay function.

**MAS-DNP sample preparation**. The stock solution of AMUPol[80] was freshly prepared in the d$_8$-glycerol/D$_2$O/H$_2$O (60/30/10 Vol%) solvent mixture referred as the DNP matrix, and a final radical concentration of 10 mM. To prepare the DNP sample, 50 μL of the stock solution was added into 5 mg of the $^{13}$C,$^{15}$N-*A. fumigatus* sample and grinded for 10–15 min to allow the radical solution to penetrate into the cell walls. 3-mg of well-hydrated samples were transferred into a 3.2-mm sapphire rotor. A 30-fold enhancement factor of NMR sensitivity with and without microwave irradiation ($\varepsilon_{on/off}$) has been achieved. Relatively short buildup times of 3–5 s indicate a good mixing of the radicals and biomolecules in these whole-cell samples. It should be noted that simply mixing the materials with the DNP matrix failed to give any enhancement regardless of the mixing time. Therefore, grinding the fungi in DNP matrices thoroughly for several minutes is needed to ensure homogeneous mixing of radicals and biomolecules in these whole-cell samples.

**MAS-DNP solid-state NMR experiments**. All the experiments were performed on a 600 MHz/395 GHz 89 mm-bore MAS-DNP spectrometer equipped with a gyrotron microwave source (National High Magnetic Field Laboratory, Tallahassee). The cathode currents of the gyrotron were 160 mA. All DNP spectra were measured using a 3.2 mm probe under 8 kHz MAS frequency. The temperature was 103.6 K with the microwave (μW) off and 106 K with the μW on.

To select chitin signals and identify chitin–glucan interactions, 2D $^{15}$N–$^{13}$C TEDOR[52] correlation experiments were implemented by a mixing period with 100-ms DARR or 3-s PDSD. Spectral subtraction generates a difference spectrum that clearly revealed seven intermolecular cross peaks. The total experimental time is 22 h. Second, to further improve the spectral resolution, we conducted a $^{15}$N,$^{13}$C filtered 2D $^{13}$C–$^{13}$C correlation experiment[23], which benefits from the presence of two $^{13}$C dimensions and provides unambiguous detection of chitin-proximal biomolecules. It took 4.5 h to measure this experiment on 3 mg $^{13}$C, $^{15}$N-fungi and 25 intermolecular cross peaks have been identified. In addition, a $^{13}$C–$^{13}$C dipolar-INADEQUATE-PDSD spectrum was measured to identify the signals of minor species and to further detect long-range correlations. To determine the packing of different chitin allomorphs, we measured 2D $^{15}$N–$^{15}$N homonuclear correlation spectra using 5 ms and 15 ms PAR mixing[46,47]. The radiofrequency field strengths for PAR were 34 kHz for $^{15}$N and 56 kHz for $^{1}$H.

**Data availability**. The data that support the findings of this study are available from the corresponding author upon request.

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

## Acknowledgements

This work was supported by Louisiana State University startup funds and LSU Biomedical Collaborative Research Program. The glycosyl composition and linkage analysis were supported in part by the Chemical Sciences, Geosciences and Biosciences Division, Office of Basic Energy Sciences, U.S. Department of Energy grant (DE-SC0015662) to CCRC at UGA. P.W. laboratory research was supported by NIH grants AI121451 and AI121460. The National High Magnetic Field Laboratory is supported by National Science Foundation through DMR-1157490 and the State of Florida. The MAS-DNP system at NHMFL is funded in part by NIH S10 OD018519 and NSF CHE-1229170. We thank Dr. Zhehong Gan and Dr. Ivan Hung for experimental assistance.

## Author contribution

X.K., A.K., F.M.-V., and T.W. designed and conducted the NMR and MAS-DNP experiments. X.K., A.K., and A.C. prepared and optimized the DNP samples. A.M. and P. A. conducted the glycosyl composition and linkage analysis. X.K., A.K., M.C.D.W., A.C.,

P.W., and T.W. analyzed the experimental data. X.K., A.K., P.W., A.M., F.M.-V., and T.W. wrote the manuscript.

## Additional information

**Competing interests:** The authors declare no competing interests.

