## [Peer Review File · Nature Communications]

REVIEWERS' COMMENTS:

Reviewer #1 (Remarks to the Author):

In this elegant work the investigators utilized a robust technology based on solid-state NMR spectroscopy, to profile the architecture of the fungal cell wall of the major human fungal pathogen *Aspergillus fumigatus*. The work is mainly focused on the analysis of *A. fumigatus* cell wall polysaccharides and provides unprecedented detail in terms of spatial resolution in a non-destructive way. Nonetheless the work is largely confirmatory of previous elegant studies utilizing polysaccharide biochemistry and mutation analysis at Pasteur Institute, Paris over the last 20 years. The main findings are (i) the interaction of chitin with α -1-3-glucans that results in a rigid and hydrophobic scaffold, which is covered by (ii) an hydrophilic layer of β -glucans that covers α -1-3-glucan-chitin layer and (iii) a highly mobile outer layer of proteins and polysaccharides mainly α -1-3-glucan.

I am concerned by the fact that the authors focused exclusively on polysaccharides to characterize cell wall architecture. Their findings are largely confirmatory of previous work. The major contribution of DHN melanin and the hydrophobin rodlet protein layer on cell wall surface hydrophobicity and rigidity is missing. Finally, the dynamics of cell wall composition during germination (cell wall swelling) are not captured in the present study

In order to make this work really novel and informative the association of DHN melanin with the polysaccharide layers in dormant and swollen conidia should be analyzed. This information has not been clearly addressed and has a fundamental interest for researchers working in the field.

In addition, the special distribution of fungal cell wall molecules in dormant vs swollen conidia of *A. fumigatus* should be reported.

Otherwise, the information provided in the manuscript is rather confirmatory and will not attract the interest of the diverse readership of Nature Communications

Reviewer #2 (Remarks to the Author):

In this work, Kang et al. have obtained a wealth of information on the architecture of fungal cell walls. Based on their results, they provide a full picture of the cell wall as composed of rigid hydrophobic chitin- α -1,3-glucan regions which are sandwiched between hydrated mobile β -glucans and an outer shell composed of highly mobile glycoproteins and α -1,3-glucans.

The English throughout the text is clear despite some minor phrasing problems or typos which the editors should easily spot. The introduction is straight to the point and introduces well the background necessary to situate this work. The literature cited is relevant and correctly referenced (except for the assignments in Supplementary table 2 for which no references are provided). The figures are clear and of high quality and judiciously chosen. The necessary more specialized or technical data can be found in the supplementary information.

The goals of the research are clearly described and of importance since the knowledge on fungal cell wall architecture is scarce. The methodology to achieve this goal is sound and state-of-the-art. The quality of the NMR data is indeed impressive considering the complexity of the cell wall. All the necessary information to replicate the experiments is provided.

The authors provide an important step forward in the understanding of fungal cell wall architecture. The data is overall solid and the findings well supported.

This is overall an excellent work and I have no comments besides the very minor typos and references. The manuscript could be published as is after these two issues are addressed.

Response to Reviewers' Comments

Reviewer #1:

In this elegant work the investigators utilized a robust technology based on solid-state NMR spectroscopy, to profile the architecture of the fungal cell wall of the major human fungal pathogen *Aspergillus fumigatus*. The work is mainly focused on the analysis of *A. fumigatus* cell wall polysaccharides and provides unprecedented detail in terms of spatial resolution in a non-destructive way.

We thank the reviewer for the positive comments regarding our study.

Nonetheless the work is largely confirmatory of previous elegant studies utilizing polysaccharide biochemistry and mutation analysis at Pasteur Institute, Paris over the last 20 years. The main findings are (i) the interaction of chitin with α -1-3-glucans that results in a rigid and hydrophobic scaffold, which is covered by (ii) an hydrophilic layer of β -glucans that covers α -1-3-glucan-chitin layer and (iii) a highly mobile outer layer of proteins and polysaccharides mainly α -1-3-glucan.

We thank the reviewer for the comments that are both critical and insightful. The knowledge gained from previous research has made it possible for us to carry out this study that results in many novel findings. First, the high-resolution data of intact fungal cells unveiled the unexpected structural polymorphism of cell wall molecules and their functional relevance. Second, this study provides consolidated and high-resolution restraints obtained from intact cells, which substantially improved our understanding of the cell walls. Third, the high-resolution method presented can be widely applied to investigate other fungi or organisms

In a review by Jean Paul Latgé (Mol. Microbiol. 2007), it is stated that "our understanding of the overall organization of all these different polysaccharides in the cell wall remains quite vague." Clearly, major issues of the cell wall structure remain unresolved and the field remains to be improved. The systematic approach presented in this study provides a possibility for resolving these issues and enables future in-situ investigations of many relevant questions that have been difficult to answer.

We have now added 12 additional references. We have also provided 14 references in the supplementary information to detail the resonance assignment.

I am concerned by the fact that the authors focused exclusively on polysaccharides to characterize cell wall architecture. Their findings are largely confirmatory of previous work. The major contribution of DHN melanin and the hydrophobin rodlet protein layer on cell wall surface hydrophobicity and rigidity is missing. Finally, the dynamics of cell wall composition during germination (cell wall swelling) are not captured in the present study.

In order to make this work really novel and informative the association of DHN melanin with the polysaccharide layers in dormant and swollen conidia should be analyzed. This information has not been clearly addressed and has a fundamental interest for researchers working in the field. In addition, the special distribution of fungal cell wall molecules in dormant vs swollen conidia

of *A. fumigatus* should be reported. Otherwise, the information provided in the manuscript is rather confirmatory and will not attract the interest of the diverse readership of Nature Communications

We thank the reviewer for the critical comments that actually point out three challenging aspects we encountered during our studies.

First, the *A. fumigatus* samples we prepared had negligible signals for the aromatic pigments, either due to its low abundance or the lack of isotope-labeled intermediates/precursors that is specially needed for isotope-labeling of these aromatic assemblies. The aromatic signals account for ~6% of the total intensity in the quantitative ^{13}C spectrum, and these aromatic signals mainly originate from protein sidechains or the overlapped lipid peaks (see the figure below). Consequently, we are unable to observe any melanin-carbohydrates correlation using the current whole-cell sample even with the assistance of DNP. Fortunately, multiple ssNMR studies on extracted melanin “ghosts” have revealed that pigments should be deposited in a carbohydrate scaffold and our results clearly revealed a stiff and hydrophobic scaffold formed by chitin and α -glucans. We have added this insight as the fifth paragraph in the Discussion section.

Second, we have indeed systematically analyzed the hydration and mobility of proteins and identified a group of proteins that are both rigid and poorly hydrated. However, we were limited by lack of the approach to distinguish proteins in the hydrophobin rodlet layer from the membrane proteins. we have presented this finding by adding a new supplementary figure Fig.S7 (See below) and Table S8-9. We have also updated Results section by adding the last paragraph.

Third, we fully agree with the reviewer that the dynamic structure is an important aspect of the fungal cell wall, however, studies involving swollen conidia remain technically challenging at the present. We hope that we will be able to assess the cell wall structure at various development stages, including swollen conidia, in future studies.

Reviewer #2:

In this work, Kang et al. have obtained a wealth of information on the architecture of fungal cell walls. Based on their results, they provide a full picture of the cell wall as composed of rigid hydrophobic chitin- α -1,3-glucan regions which are sandwiched between hydrated mobile β -glucans and an outer shell composed of highly mobile glycoproteins and α -1,3-glucans.

We thank the reviewer for the positive comments regarding our study.

The English throughout the text is clear despite some minor phrasing problems or typos which the editors should easily spot.

We have addressed this issue in the revised manuscript.

The introduction is straight to the point and introduces well the background necessary to situate this work. The literature cited is relevant and correctly referenced (except for the assignments in Supplementary table 2 for which no references are provided).

We have now added two new columns to Supplementary Table 2 (see the snapshot below) including 14 references and our experimental methods for the resonance assignment.

	C1	C2	C3	C4	C5	C6	CO	CH ₃	N	Experimental methods	References
β -1,3-glucan ^a	103.6	74.4	86.4	68.7	77.1	61.3	/	/	/	¹³ C, ¹³ C PDS, ¹³ C CP J-INADEQUATE	Shim et al. 2007
β -1,3-glucan ^b (w)	104.6	75.2	85.9	69.6	78.6	63.0	/	/	/	DNP ¹³ C, ¹³ C dipolar-INADEQUATE-PDS	Fairweather et al. 2004 Hazime Saitô et al. 1979¹⁻³
β -1,3-glucan ^c (w)	102.6	72.8	84.4	69.7	72.2	60.5	/	/	/	DNP ¹³ C CP J-INADEQUATE	
α -1,3-glucan ^a	101.0	71.9	84.6	69.5	71.7	60.5	/	/	/	¹³ C, ¹³ C PDS, ¹³ C CP J-INADEQUATE	Bhanja et al. 2014⁴ Puanglek et al. 2016⁵
α -1,3-glucan ^b	99.9	71	80	70.4	69.7	60.9	/	/	/		
α -1,3-glucan ^c	101.2	70.1	84.5	67.7	71.5	60.5	/	/	/		

The figures are clear and of high quality and judiciously chosen. The necessary more specialized or technical data can be found in the supplementary information. The goals of the research are clearly described and of importance since the knowledge on fungal cell wall architecture is scarce. The methodology to achieve this goal is sound and state-of-the-art. The quality of the NMR data is indeed impressive considering the complexity of the cell wall. All the necessary information to replicate the experiments is provided. The authors provide an important step forward in the understanding of fungal cell wall architecture. The data is overall solid and the findings well supported.

We thank the reviewer for acknowledging the impact and soundness of our work.

This is overall an excellent work and I have no comments besides the very minor typos and references. The manuscript could be published as is after these two issues are addressed.

We have addressed these issues in the revised manuscript as detailed above.